Corrected: Author correction; Author correction

# Genome-wide association study of alcohol consumption and use disorder in 274,424 individuals from multiple populations

Henry R. Kranzler [1,2], Hang Zhou [3,4], Rachel L. Kember[1,2], Rachel Vickers Smith [2,5], Amy C. Justice [3,4,6], Scott Damrauer[1,2], Philip S. Tsao[7,8], Derek Klarin [9], Aris Baras[10], Jeffrey Reid [10], John Overton[10], Daniel J. Rader[1], Zhongshan Cheng[3,4], Janet P. Tate[3,4], William C. Becker[3,4], John Concato[3,4], Ke Xu[3,4], Renato Polimanti[3,4], Hongyu Zhao [3,6] & Joel Gelernter [3,4]

Alcohol consumption level and alcohol use disorder (AUD) diagnosis are moderately heritable traits. We conduct genome-wide association studies of these traits using longitudinal Alcohol Use Disorder Identification Test-Consumption (AUDIT-C) scores and AUD diagnoses in a multi-ancestry Million Veteran Program sample ($N = 274,424$). We identify 18 genome-wide significant loci: 5 associated with both traits, 8 associated with AUDIT-C only, and 5 associated with AUD diagnosis only. Polygenic Risk Scores (PRS) for both traits are associated with alcohol-related disorders in two independent samples. Although a significant genetic correlation reflects the overlap between the traits, genetic correlations for 188 non-alcohol-related traits differ significantly for the two traits, as do the phenotypes associated with the traits' PRS. Cell type group partitioning heritability enrichment analyses also differentiate the two traits. We conclude that, although heavy drinking is a key risk factor for AUD, it is not a sufficient cause of the disorder.

[1] University of Pennsylvania Perelman School of Medicine, Philadelphia, PA 19104, USA. [2] Crescenz Veterans Affairs Medical Center, Philadelphia, PA 19104, USA. [3] Yale School of Medicine, New Haven, CT 06511, USA. [4] Veterans Affairs Connecticut Healthcare System, West Haven, CT 06516, USA. [5] University of Louisville School of Nursing, Louisville, KY 40202, USA. [6] Yale School of Public Health, New Haven, CT 06511, USA. [7] VA Palo Alto Health Care System, Palo Alto, CA 94304, USA. [8] Stanford University School of Medicine, Stanford, CA 94305, USA. [9] Massachusetts General Hospital, Harvard Medical School, Boston, MA 02114, USA. [10] Regeneron Genetics Center, Tarrytown, NY 10591, USA. These authors contributed equally: Henry R. Kranzler, Hang Zhou, Rachel L. Kember. Correspondence and requests for materials should be addressed to H.R.K. (email: kranzler@pennmedicine.upenn.edu)

Excessive alcohol consumption is associated with a host of adverse medical, psychiatric, and social consequences. Globally, in 2012, about 3.3 million or 5.9% of all deaths, 139 million disability-adjusted life years, and 5.1% of the burden of disease and injury were attributable to alcohol consumption, with the magnitude of harm determined by the volume of alcohol consumed and the drinking pattern[1]. Regular heavy drinking is the major risk factor for the development of an alcohol use disorder (AUD), a chronic, relapsing condition characterized by impaired control over drinking[2]. Independent of AUD, heavy drinking has a multitude of adverse medical consequences. Identifying factors that contribute to drinking level and AUD risk could advance efforts to prevent, identify, and treat both medical and psychiatric problems related to alcohol.

Many different alcohol-related phenotypes have been used to investigate genetic risk, including formal diagnoses, such as alcohol dependence [e.g., based on the Diagnostic and Statistical Manual of Mental Disorders, 4th edition (DSM-IV)[3]] and screening tests that measure alcohol consumption and alcohol-related problems [e.g., the Alcohol Use Disorders Identification Test (AUDIT)]. The AUDIT, a 10-item, self-reported test developed by the World Health Organization as a screen for hazardous and harmful drinking[4,5] has been used for genome-wide association studies (GWASs) both as a total score[6–8] and as the AUDIT-Consumption (AUDIT-C) and AUDIT-Problems (AUDIT-P) sub-scores[8]. The three-item AUDIT-C measures the frequency and quantity of usual drinking and the frequency of binge drinking, while the 7-item AUDIT-P measures alcohol-related problems.

Twin and adoption studies have shown that half of the risk of alcohol dependence, a subtype of AUD, is heritable[9]. The single-nucleotide polymorphism (SNP) heritability of alcohol dependence in a family-based, European-American (EA) sample was 16%[10] and 22% in an unrelated African-American (AA) sample[11]. In the meta-analysis of data from the UK Biobank (UKBB) and 23andMe, the SNP heritability of the total AUDIT was estimated to be 12%, while for the AUDIT-C and AUDIT-P it was 11% and 9%, respectively).

In 12 GWASs of alcohol dependence (most of which used a binary DSM-IV diagnosis[3]) published between 2009 and 2014 (ref. [12]), the only consistent genome-wide significant (GWS) findings were for SNPs in genes encoding the alcohol metabolizing enzymes. Similarly, in a recent meta-analysis of 14,904 individuals with alcohol dependence and 37,944 controls, which was stratified by genetic ancestry (European, $N = 46,568$; African; $N = 6280$), the only GWS findings were two independent *ADH1B* variants. In addition, there were significant genetic correlations seen with 17 phenotypes, including psychiatric (e.g., schizophrenia, depression), substance use (e.g., smoking and cannabis use), social (e.g., socio-economic deprivation), and behavioral (e.g., educational attainment) traits[13].

Alcohol-metabolizing enzyme genes have also been associated with mean or maximal alcohol consumption levels, potential intermediate phenotypes for alcohol dependence[14–19]. In a meta-analysis of GWASs ($N > 105,000$ European subjects), *KLB* was associated with alcohol consumption[20]. A GWAS of alcohol consumption in the UK Biobank sample[21] identified GWS associations at 14 loci (8 independent), including three alcohol-metabolizing genes on chromosome 4 (*ADH1B*, *ADH1C*, and *ADH5*), an intergenic SNP on chromosome 4, and *KLB*, replicating the prior meta-analytic findings. Risk genes identified in this study included *GCKR*, *CADM2*, and *FAM69C*.

A GWAS of the AUDIT in nearly 8000 individuals failed to identify any GWS loci[6]. A GWAS of the AUDIT from 23andMe in 20,328 European ancestry participants also failed to yield GWS results[7], although meta-analysis of the AUDIT in the UKBB and

23andMe samples identified 10 associated risk loci, including associations to *JCAD* and *SLC39A13* (ref. [8]). In addition to the total AUDIT-C score, the meta-analysis included GWASs for the AUDIT-C and AUDIT-P, which showed significantly different patterns of association across a number of traits, including psychiatric disorders. Specifically, the direction of genetic correlations between schizophrenia, major depressive disorder, and obesity (among others) was negative for AUDIT-C and positive for AUDIT-P.

In the present study, we evaluate the independent and overlapping genetic contributions to AUDIT-C and AUD in a single large multi-ancestry sample from the Million Veteran Program (MVP)[22]. Large-scale biobanks such as the MVP offer the potential to link genes to health-related traits documented in the electronic health record (EHR) with greater statistical power than can ordinarily be achieved in prospective studies[23]. Such discoveries improve our understanding of the etiology and pathophysiology of complex diseases and their prevention and treatment. To that end, we use a common data source—longitudinal repeated measures of alcohol-related traits from the national Veterans Health Administration (VHA) EHR—to obtain the mean, age-adjusted AUDIT-C score and International Classification of Diseases (ICD) alcohol-related diagnosis codes over more than 11 years of care[24]. We then conduct a GWAS of each trait followed by downstream analysis of the findings in which we construct Polygenic Risk Scores (PRS) for both traits and show that they are associated with alcohol-related disorders in two independent samples. The availability of data on alcohol consumption from the AUDIT-C and a formal diagnosis of AUD from the EHR enables us to examine the relationship between these key alcohol-related traits in a single, well-phenotyped sample and to compare the findings for these traits more systematically than has previously been possible.

## Results

**Principal components analysis.** We differentiated participants genetically into five populations (see Methods, Supplementary Fig. 1) and removed outliers. There was a high degree of concordance (Supplementary Fig. 2) between the genetically defined populations and the self-reported groups for European Americans (EAs, 95.6% were self-reported Non-Hispanic white) and African Americans (AAs, 94.5% were self-reported Non-Hispanic black). Concordance ranged from 53.1% to 81.6% in the other three population groups.

**GWAS analyses.** The GWAS for AUDIT-C (Fig. 1a, Table 1 and Supplementary Table 1 and Supplementary Data 1) identified 13 independent loci in EAs, 2 in AAs, 1 in LAs (Hispanic and Latino Americans), and 1 in EAAs (East Asian Americans) (Supplementary Figs. 4, 5). Meta-analysis across the five populations (see Methods) also yielded 13 independent loci, 5 of which were previously associated with a self-reported measure of alcohol consumption: *GCKR*[21], *KLB*[20,21], *ADH1B*[18,21], *ADH1C*[21], and *SLC39A8* (ref. [8]). The eight trans-population signals for AUDIT-C identified here include *VRK2* (Vaccinia related kinase 2), *DCLK2* (Doublecortin like kinase 2), *ISL1* (ISL LIM Homeobox 1), *FTO* (Alpha-Ketoglutarate Dependent Dioxygenase), *IGF2BP1* (Insulin like growth factor 2 MRNA binding protein 1), *PPR1R3B* (Protein phosphatase 1 regulatory subunit 3B), *BRAP* (BRCA1 associated protein), *BAHCC1* (BAH domain and coiled-coil containing 1), and *RBX1* (Ring-box 1). *BAHCC1* and *RBX1* were GWS only in the trans-population meta-analysis, the results of which were driven largely by the findings in EAs, who comprised 73.5% of the total MVP sample.

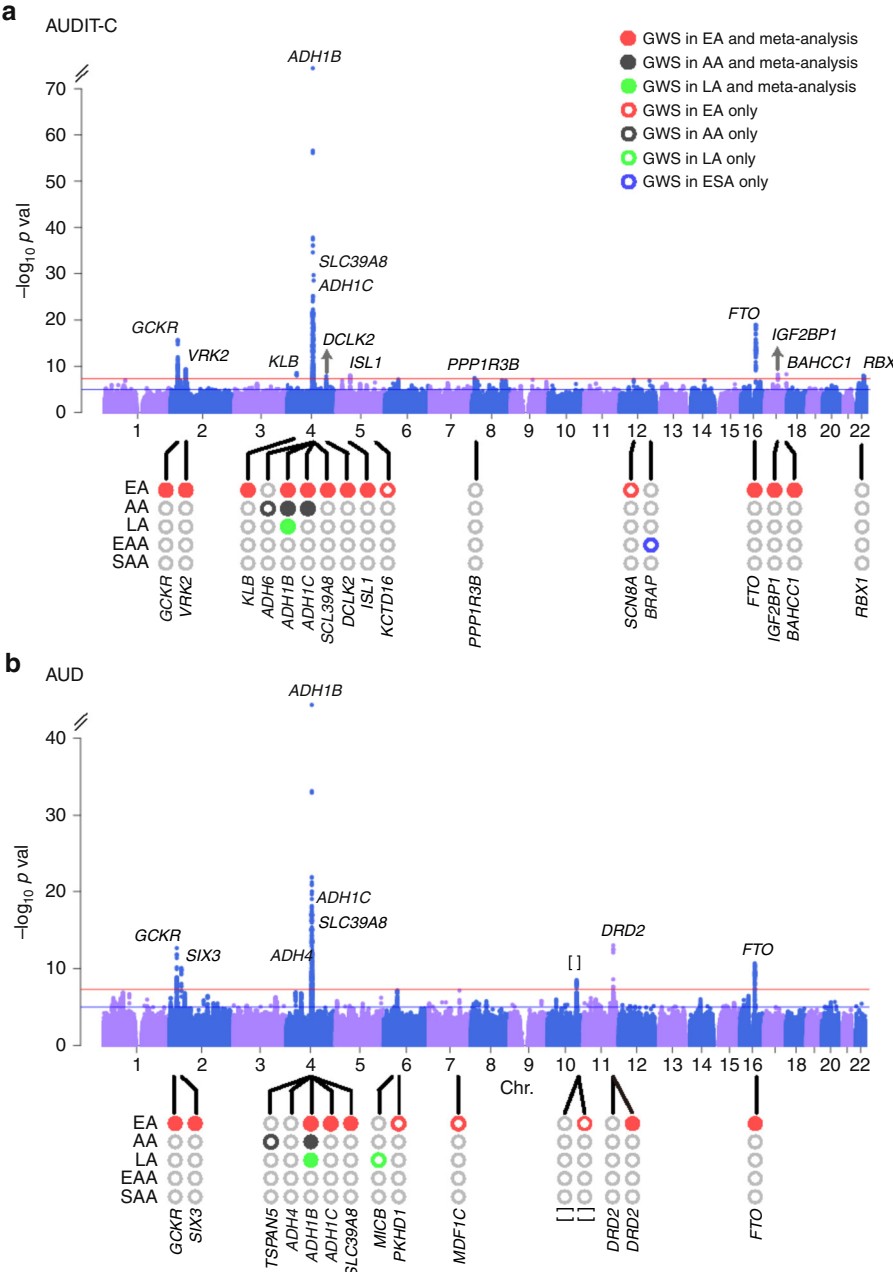

**Fig. 1** Manhattan plots for age-adjusted mean AUDIT-C score and AUD diagnosis. **a** Manhattan plot of the genome-wide association meta-analysis of AUDIT-C across all five populations ($N = 272,842$). **b** Manhattan plot of the genome-wide association meta-analysis of AUD across five populations (55,584 cases and 218,807 controls). Red lines show the genome-wide significance level ($5.0 \times 10^{-8}$). EA: European American, AA: African American, LA: Hispanic or Latino, EAA: East Asian American, SAA: South Asian American. Labeled genes at the top of the peaks indicate completely independent signals after conditional analysis in meta-analysis. Population-specific loci are labeled at the bottom of the circles in the lower part of each figure. []: no genes within 500 kb to the lead SNP

The GWAS for AUD (Fig. 1b, Table 2 and Supplementary Tables 1 and Supplementary Data 2) identified 10 independent loci in EAs, 2 in AAs, and 2 in LAs (Supplementary Figs. 6, 7). Meta-analysis across the five populations yielded 10 independent loci, including 3 previously associated with alcohol dependence[25] —ADH1B, ADH1C, and ADH4—and 7 loci not previously associated with an AUD diagnosis: GCKR, SIX3 (SIX Homeobox 3), SLC39A8, DRD2 (Dopamine Receptor D2: rs4936277 and rs61902812, which were independent), chr10q25.1 (rs7906104), and FTO. Five loci were significant in both the AUDIT-C and AUD GWASs (Supplementary Fig. 8): ADH1B, ADH1C, FTO, GCKR, and SLC39A8. The trans-population GWS findings for AUD are also driven largely by the findings in EAs.

The GWAS findings largely reflect male-specific signals due to the predominantly male sample (Supplementary Table 1). However, sex-stratified GWAS also identified two female-specific signals for AUDIT-C (Supplementary Data 3, Supplementary Figs. 9, 10) and one for AUD (Supplementary Data 4, Supplementary Figs. 11, 12).

For AUDIT-C, when associations for the seven LD-pruned GWS SNPs on chromosome 4q23–q24 in EAs are conditioned on rs1229984, the most significant functional SNP in the region in that population, the only independent signal (using a Bonferroni-corrected $p$ value < 0.05) is for rs1229978, near ADH1C (Supplementary Data 5). For AUD, when associations for the four LD-pruned GWS SNPs in EAs are conditioned on

**Table 1 Genome-wide significant associations for AUDIT-C in the trans-population meta-analysis**

| rsID | Chr:pos[a] | A1/A2 | Gene[b] | EAF | N | Z-score | P_EA | P_AA | P_LA | P_EAA | P_SAA | Effect | P_meta |
|------|-----------|-------|---------|-----|---|---------|------|------|------|-------|-------|--------|--------|
| rs1260326 | 2:27730940 | C/T | GCKR[c] | 0.652 | 270,226 | 8.22 | $1.74 \times 10^{-16}$ | 0.110 | 0.067 | 0.987 | 0.739 | +++++ | $2.04 \times 10^{-16}$ |
| rs2683616 | 2:58035555 | A/G | VRK2[d] | 0.624 | 211,399 | 6.22 | $1.80 \times 10^{-9}$ | NA | 0.060 | 0.487 | NA | +?+−? | $4.95 \times 10^{-10}$ |
| rs12639940 | 4:39420981 | A/G | KLB[c] | 0.613 | 194,761 | 5.93 | $3.45 \times 10^{-9}$ | NA | NA | 0.626 | NA | +??+? | $3.06 \times 10^{-9}$ |
| rs1229984 | 4:100239319 | C/T | ADH1B[c] | 0.970 | 272,358 | 24.56 | $4.83 \times 10^{-102}$ | $1.31 \times 10^{-19}$ | $4.40 \times 10^{-16}$ | $9.05 \times 10^{-3}$ | NA | ++++? | $3.62 \times 10^{-133}$ |
| rs142783062 | 4:100270960 | D/I | ADH1C[c] | 0.345 | 271,444 | 9.82 | $2.04 \times 10^{-14}$ | $4.75 \times 10^{-7}$ | $4.90 \times 10^{-4}$ | 0.019 | 0.779 | +++++ | $9.50 \times 10^{-23}$ |
| rs13107325 | 4:103188709 | C/T | SLC39A8[c] | 0.937 | 270,248 | 11.45 | $1.43 \times 10^{-25}$ | $1.07 \times 10^{-4}$ | $2.23 \times 10^{-3}$ | NA | NA | +++?? | $2.24 \times 10^{-30}$ |
| rs4423856 | 4:150984857 | T/C | DCLK2[d] | 0.796 | 212,444 | 5.66 | $3.60 \times 10^{-8}$ | NA | 0.289 | 0.574 | 0.144 | +?+++ | $1.48 \times 10^{-8}$ |
| rs2961816 | 5:50443691 | A/C | ISL1[d] | 0.683 | 260,828 | 5.74 | $1.24 \times 10^{-7}$ | 0.021 | 0.641 | 0.932 | 0.137 | +++++ | $9.75 \times 10^{-9}$ |
| rs4841132 | 8:9183596 | A/G | PPP1R3B[d] | 0.101 | 271,192 | −5.51 | $2.75 \times 10^{-6}$ | 0.226 | NA | NA | 0.148 | −−−?+ | $3.62 \times 10^{-8}$ |
| rs62033408 | 16:53827962 | A/G | FTO[c] | 0.678 | 270,067 | 9.08 | $2.20 \times 10^{-15}$ | $4.78 \times 10^{-5}$ | 0.229 | 0.177 | 0.027 | +++++ | $1.11 \times 10^{-19}$ |
| rs9902512 | 17:47094274 | C/G | IGF2BP1[c] | 0.664 | 207,229 | −5.81 | $3.81 \times 10^{-8}$ | NA | 0.055 | 0.782 | NA | −?−−? | $6.24 \times 10^{-9}$ |
| rs142997686 | 17:79419159 | D/I | BAHCC1[c] | 0.384 | 211,399 | 5.84 | $1.77 \times 10^{-9}$ | NA | 0.944 | 0.434 | 0.840 | +?+−+ | $5.39 \times 10^{-9}$ |
| rs75723348 | 22:41420679 | T/G | RBX1[d] | 0.736 | 269,785 | 5.71 | $2.97 \times 10^{-7}$ | 0.063 | 0.072 | 0.536 | 0.579 | +++++ | $1.11 \times 10^{-8}$ |

The loci shown represent completely independent signals after conditioning analyses
A1 effect allele, A2 other allele, EAF effective allele frequency, EA European American, AA African American, LA Latino American, EAA East Asian American, SAA South Asian American
[a]Human Genome hg19 assembly
[b]Gene nearest to the lead SNP
[c]Protein-coding gene contains the lead SNP
[d]Protein-coding gene nearest to the lead SNP

**Table 2 Genome-wide significant associations for AUD in the trans-population meta-analysis**

| rsID | Chr:pos[a] | A1/A2 | Gene[b] | EAF | N | Z-score | P_EA | P_AA | P_LA | P_EAA | P_SAA | Effect | P_meta |
|------|-----------|-------|---------|-----|---|---------|------|------|------|-------|-------|--------|--------|
| rs1260326 | 2:27730940 | C/T | GCKR[c] | 0.651 | 271,763 | 7.33 | $1.44 \times 10^{-16}$ | 0.679 | 0.778 | 0.830 | 0.820 | +++−+ | $2.27 \times 10^{-13}$ |
| rs540606 | 2:45138507 | A/G | SIX3[d] | 0.409 | 213,336 | −6.49 | $2.84 \times 10^{-10}$ | 0.175 | 0.411 | NA | 0.412 | −?−−? | $8.58 \times 10^{-11}$ |
| rs5860563 | 4:100047157 | D/I | ADH4[c] | 0.723 | 271,487 | −6.09 | $7.63 \times 10^{-5}$ | $9.85 \times 10^{-7}$ | 0.035 | NA | 0.412 | −−−? | $1.12 \times 10^{-9}$ |
| rs1229984 | 4:100239319 | C/T | ADH1B[c] | 0.969 | 273,904 | 19.54 | $4.51 \times 10^{-74}$ | $4.18 \times 10^{-5}$ | $5.81 \times 10^{-17}$ | 0.032 | NA | ++++? | $4.68 \times 10^{-85}$ |
| rs1612735 | 4:100258007 | T/C | ADH1C[c] | 0.656 | 271,471 | −8.86 | $1.75 \times 10^{-14}$ | $6.42 \times 10^{-5}$ | 0.022 | 0.938 | 0.054 | −−−++ | $7.90 \times 10^{-19}$ |
| rs13107325 | 4:103188709 | C/T | SLC39A8[c] | 0.937 | 271,784 | 7.60 | $2.73 \times 10^{-14}$ | 0.064 | 0.363 | NA | NA | +++?? | $2.97 \times 10^{-14}$ |
| rs7906104 | 10:110497101 | T/C | | 0.272 | 270,278 | −5.92 | $3.15 \times 10^{-7}$ | $8.72 \times 10^{-3}$ | 0.357 | 0.195 | 0.106 | −−−−− | $3.17 \times 10^{-9}$ |
| rs61902812 | 11:113374420 | A/C | DRD2[d] | 0.304 | 271,218 | −5.58 | $4.99 \times 10^{-6}$ | 0.025 | 0.015 | 0.220 | 0.931 | −−−−− | $2.44 \times 10^{-8}$ |
| rs4936277[e] | 11:113431960 | A/G | DRD2[d] | 0.599 | 274,128 | 7.44 | $2.85 \times 10^{-11}$ | 0.073 | $4.36 \times 10^{-4}$ | 0.200 | 0.357 | +++++ | $1.01 \times 10^{-13}$ |
| rs1421085 | 16:53800954 | T/C | FTO[c] | 0.670 | 274,340 | 6.69 | $3.26 \times 10^{-10}$ | 0.024 | 0.525 | 0.332 | 0.018 | +++++ | $2.17 \times 10^{-11}$ |

The loci shown represent completely independent signals after conditioning analyses
A1 effect allele, A2 other allele, EAF effective allele frequency, EA European American, AA African American, LA Latino American, EAA East Asian American, SAA South Asian Americans
[a]Human Genome hg19 assembly
[b]Gene nearest to the lead SNP
[c]Protein-coding gene contains the lead SNP
[d]Protein-coding gene nearest to the lead SNP
[e]Different signal than rs61902812

rs1229984, the only independent signal in that region is for rs1154433 near *ADH1C*. In the trans-population meta-analysis, rs5860563 is independent when conditioned on rs1229984 in EAs, and on rs2066702 in AAs, the most significant functional SNP in the region in AAs (Supplementary Data 6).

To elucidate further the genetic differences between AUDIT-C and AUD, we conducted a GWAS of each phenotype with the other phenotype as a covariate. A GWAS of AUDIT-C with AUD as a covariate identified 10 GWS loci in EAs and 2 GWS loci in AAs (Supplementary Data 7). In both EAs and AAs, all loci overlapped with the GWS findings for AUDIT-C alone. A GWAS of AUD that included AUDIT-C as a covariate identified five GWS loci in EAs and one in AAs (Supplementary Data 8). Among EAs, four of the loci were the same as for AUD, the only non-overlapping finding being *DIO1* (Iodothyronine Deiodinase 1). In AAs, *ADH1B* remained significant for AUD when accounting for AUDIT-C, but *TSPAN5* did not.

Using a sign test, most SNPs have the same direction of effect for AUDIT-C and AUD, consistent with the high genetic correlation between the traits. For SNPs with $p$ value $<1 \times 10^{-6}$ the sign concordance between the two traits is 98.7% in EAs and 100% in the other four, smaller population groups.

**Body mass index-adjusted GWAS.** Because *FTO* was GWS for both AUDIT-C and AUD, we repeated the two GWASs correcting for body mass index (BMI). Among the top SNPs associated with AUDIT-C and AUD, most remain GWS after correction for BMI, though the significance level of some change (Supplementary Data 9, 10). *FTO* SNPs become only nominally significant for both alcohol-related traits: the $p$ value for the lead SNP for AUDIT-C, rs9937709, decreases in significance from $5.53 \times 10^{-14}$ to $1.42 \times 10^{-5}$ and for the lead SNP for AUD, rs11075992, it decreases in significance from $3.22 \times 10^{-10}$ to $3.02 \times 10^{-5}$. In contrast, with correction for BMI, some signals increase, e.g., for rs1260326 in *GCKR* the $p$ value increases in significance from $p = 2.04 \times 10^{-16}$ to $p = 2.91 \times 10^{-19}$ for AUDIT-C and from $p = 2.27 \times 10^{-13}$ to $p = 1.71 \times 10^{-14}$ for AUD. Similarly, rs1229984 in *ADH1B* increases in significance from $p = 3.62 \times 10^{-133}$ to $p = 9.81 \times 10^{-145}$ for AUDIT-C and from $p = 4.68 \times 10^{-85}$ to $p = 3.85 \times 10^{-89}$ for AUD.

**Gene-based analyses.** For AUDIT-C score, gene-based association analyses identify 31 genes in EAs that are GWS ($p < 2.69 \times 10^{-6}$), 3 in AAs, 1 in LAs, and 2 in EAAs (Supplementary Fig. 13), including many of the loci in the SNP-based analyses for that trait. The unique genes in EAs include *C4orf17*, *ZNF512*, *MTTP*, *TBCK*, and *MCC*. For AUDIT-C, the loci that were not GWS in the SNP-based analyses included *EIF4E* in AAs, *MAP2* in LAs, and *LOX* and *MYL2* in EAAs.

For AUD, we identify 23 GWS genes in EAs, 5 in AAs, and 1 in LAs (Supplementary Fig. 14), many of which are GWS loci in the SNP-based analyses for that trait. For AUD, the loci in EAs that are not GWS in the SNP-based analyses are *KRTCAP3*, *TRMT10A*, *ZNF512*, *DCLK2*, *MTTP*, and *MCC*. In AAs, *EIF4E*, *ADH4*, and *METAP1* are GWS for AUD, while *ADGRB2* is the only GWS locus in LAs.

**Pathway and biological enrichment analyses.** Using Functional Mapping and Annotation (FUMA)[26] software to investigate the pathway or biological process enrichment with summary statistics as input and false discovery rate (FDR) correction for multiple testing, we find multiple reactome and Kyoto Encyclopedia of Genes and Genomes (KEGG) pathways that are significantly enriched for AUDIT-C (Supplementary Data 11, Supplementary Fig. 15) and AUD (Supplementary Data 12, Supplementary Fig. 16) in each population. The most significant pathway is reactome ethanol oxidation for both traits in both EAs and AAs. Multiple GO biological processes are enriched for AUDIT-C (Supplementary Data 13, Supplementary Fig. 17) and AUD (Supplementary Data 14, Supplementary Fig. 18), including ethanol and alcohol metabolism. Enrichments for chemical and genetic perturbation gene sets and for the GWAS catalog for both traits are shown in Supplementary Data 15–18 and Supplementary Figs. 19–22.

**Heritability estimates.** We use linkage disequilibrium score regression (LDSC)[27] (see Methods) to estimate SNP-based heritability ($h_{2 \, SNP}$) in EAs and AAs, where sample sizes are large enough to provide robust estimates for each trait (Fig. 2a). For AUDIT-C, the $h_{2 \, SNP}$ is 0.068 (s.e. = 0.005) in EAs: 0.068 (s.e. = 0.005) in males and 0.099 (s.e. = 0.037) in females. In AAs, the $h_{2 \, SNP}$ is 0.062 (s.e. = 0.016): 0.058 (s.e. = 0.018) in males. For AUD, the $h_{2 \, SNP}$ is 0.056 (s.e. = 0.004) in EAs: 0.054 (s.e. = 0.004) in males and 0.110 (s.e. = 0.038) in females. The $h_{2 \, SNP}$ for AUD is

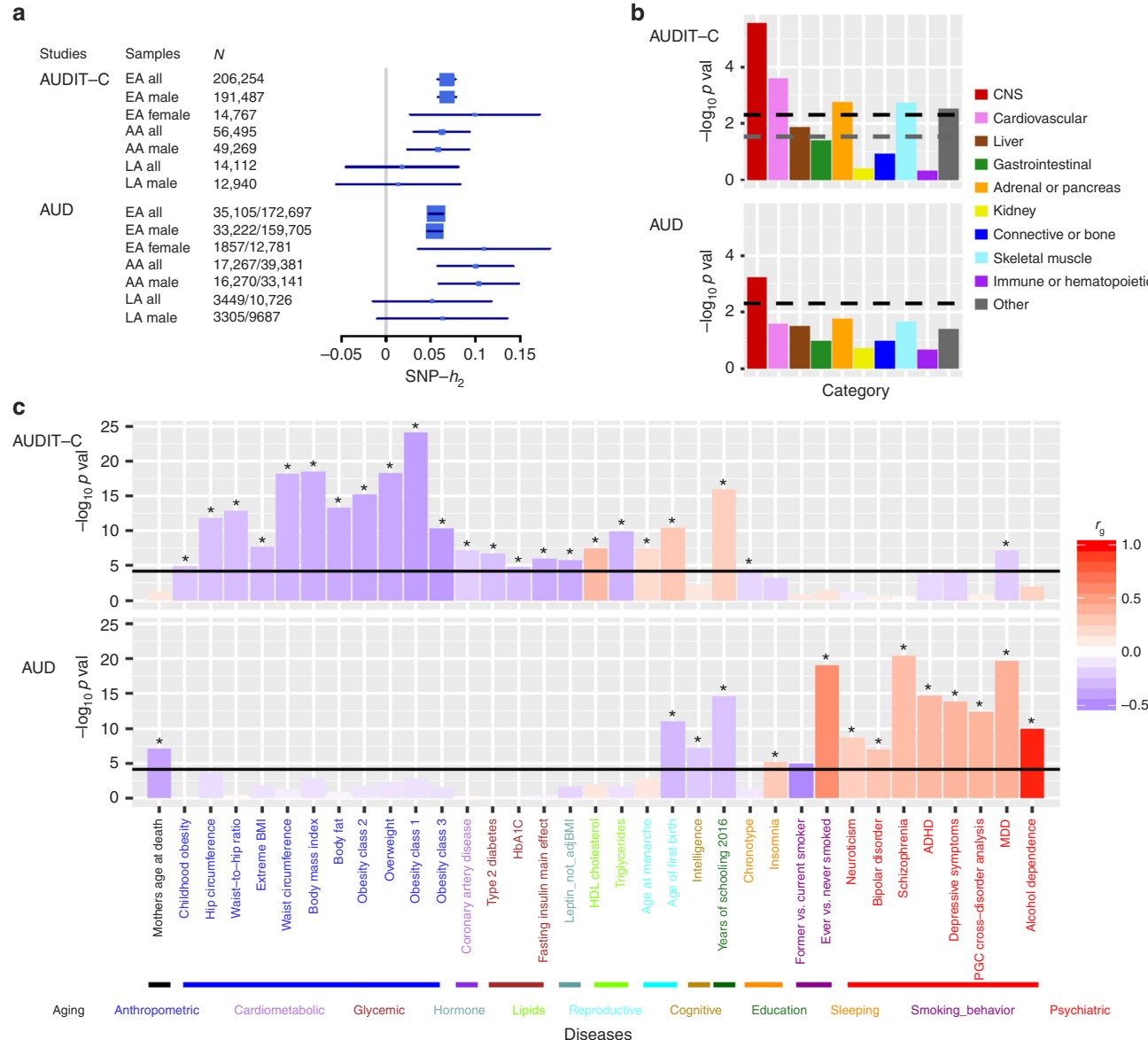

**Fig. 2** Heritability estimate, partitioning enrichments of heritability, and genetic correlation analyses using LD score regression. **a** SNP-based heritability for AUDIT-C and AUD in the three populations and sex-stratified samples adequate in size for the analysis. **b** Partitioned heritability enrichment of cell type groups for AUDIT-C and AUD. Ten cell types tested were corrected for multiple testing. The black dashed line is the cutoff for Bonferroni-corrected significance. The gray dashed line is the cutoff for FDR < 0.05. **c** Genetic correlations with other traits. Data from 714 publicly available datasets (221 published and 493 unpublished from UK Biobank) were tested and corrected for multiple comparisons. The significantly correlated traits presented are for published data. Black lines are the cutoff for Bonferroni-corrected significance, with asterisks showing traits significant after correction. The traits are grouped into different categories and sorted by the genetic correlations with AUDIT-C (upper panel) or AUD (lower panel). CNS central nervous system, ADHD attention deficit hyperactivity disorder, MDD major depressive disorder

0.100 (s.e. = 0.022) in AAs: 0.104 (s.e. = 0.023) in males. Robust estimates of $h_{2\,SNP}$ are unavailable in AA and LA females due to the small sample size.

In the analysis of stratified heritability enrichment using LDSC[28] (see Methods), several cell line functional enrichments were significant (FDR < 0.05) for AUDIT-C (Supplementary Data 19) and AUD (Supplementary Data 20). Cell type group partitioning heritability enrichment analyses indicated that central nervous system (CNS) was the most significant cell type for AUDIT-C (Fig. 2b, upper panel; Supplementary Data 21) and the only significant cell type for AUD (Fig. 2b, bottom panel; Supplementary Data 22). Enrichments for AUDIT-C were also detected for cardiovascular, adrenal or pancreatic, skeletal muscle, other, and liver cell types in descending order of significance. We also tested the heritability enrichments using data from gene expression and chromatin to identity disease-related tissues or cell types[29] (Supplementary Data 23–32). We found a few epigenetic features in brain tissues—e.g., H3K4me1, H3K4me3, and DNase—that were significantly enriched for each trait.

**Genetic correlations**. We estimated the genetic correlation ($r_g$) between different datasets or populations using LDSC[30]. The $r_g$ between AUDIT-C and AUD was 0.522 (s.e. = 0.038, $p = 2.40 \times 10^{-42}$) in EAs and 0.930 (s.e. = 0.122, $p = 1.85 \times 10^{-14}$) in AAs (Supplementary Data 33). The $r_g$ between EA males and EA females was 0.815 (s.e. = 0.156, $p = 1.69 \times 10^{-7}$) for AUDIT-C and 0.833 (s.e. = 0.142, $p = 4.16 \times 10^{-9}$) for AUD.

After Bonferroni correction, 179 traits or diseases were genetically correlated with AUDIT-C (Fig. 2c, upper panel; Supplementary Data 34). AUDIT-C was positively genetically correlated with lipids (e.g., HDL cholesterol concentration: $r_g = 0.361$, $p = 3.39 \times 10^{-8}$), reproductive traits (e.g., age at menarche: 0.190, $p = 4.20 \times 10^{-8}$), and years of education ($r_g = 0.248$, $p = 1.40 \times 10^{-16}$) and negatively correlated with anthropometric (e.g., BMI: $r_g = -0.350$, $p = 3.25 \times 10^{-19}$), cardiometabolic (e.g., coronary artery disease: $r_g = -0.212$, $p = 8.28 \times 10^{-8}$), glycemic (e.g., Type 2 diabetes: $r_g = -0.273$, $p = 2.34 \times 10^{-7}$), lipid (e.g., triglyceride concentration: $r_g = -0.325$, $p = 1.29 \times 10^{-10}$), and psychiatric (e.g., major depressive disorder (MDD) ($r_g = -0.216$, $p = 7.72 \times 10^{-8}$) traits. After correction, 111 traits or diseases were genetically associated with AUD (Fig. 2c bottom panel; Supplementary Data 35), including positive genetic correlations with sleep disturbance (e.g., insomnia: $r_g = 0.280$, $p = 7.43 \times 10^{-6}$), ever having smoked ($r_g = 0.581$, $p = 9.19 \times 10^{-20}$), and multiple psychiatric disorders (e.g., alcohol dependence: $r_g = 0.965$, $p = 1.21 \times 10^{-10}$; MDD: $r_g = 0.406$, $p = 2.19 \times 10^{-20}$), and negative genetic correlations with aging-related factors (e.g., mother's age at death: $r_g = -0.390$, $p = 8.09 \times 10^{-8}$), intelligence ($r_g = -0.226$, $p = 6.79 \times 10^{-8}$), years of education ($r_g = -0.263$, $p = 2.88 \times 10^{-15}$), and quitting smoking ($r_g = -0.517$, $p = 1.12 \times 10^{-5}$).

We tested the difference between genetic correlations for AUDIT-C and AUD using a two-tailed z-test. After correction for 714 tested traits, the genetic correlations for 188 traits showed significant differences between the two alcohol-related traits (Supplementary Data 36). We explored trait and disease associations for AUDIT-C-adjusted for AUD and AUD-adjusted for AUDIT-C, and found that the genetic correlations between the alcohol-related traits and other phenotypes did not differ substantially from the unadjusted ones (Supplementary Data 37, 38). Additionally, we explored genetic correlations for AUDIT-C-adjusted for BMI (Supplementary Data 39) and AUD-adjusted for BMI (Supplementary Data 40). Most of the genetic correlations for AUDIT-C-adjusted for BMI did not differ substantially from the unadjusted ones, except for anthropometric traits, where the negative correlation was attenuated (although still significant). Significant genetic correlations for AUD-adjusted for BMI did not differ substantially from those for AUD alone. We also explored prior GWAS associations for the GWS SNPs from AUDIT-C and AUD analyses and found associations with other phenotypes for five of them (Supplementary Data 41).

**Polygenic Risk Scores**. We examined PRS generated from the AUDIT-C and AUD GWASs in three samples (Supplementary Figs. 23–26). First, in a hold-out MVP sample of EAs and AAs (described in Methods), AUDIT-C and AUD PRS were significantly associated with both AUDIT-C and AUD phenotypes (Supplementary Data 42, 43). Lower $p$ value thresholds of AUDIT-C PRS were associated with AUDIT-C score and AUD diagnosis codes, with the most significant being $1 \times 10^{-7}$ (EA AUDIT-C: $\beta = 0.088$, $p = 1.43 \times 10^{-44}$; EA AUD: $\beta = 0.137$, $p = 3.03 \times 10^{-30}$; AA AUDIT-C: $\beta = 0.094$, $p = 2.82 \times 10^{-17}$; AA AUD: $\beta = 0.110$, $p = 1.3 \times 10^{-10}$). All $p$ value thresholds for AUD PRS were associated with both AUDIT-C score and AUD diagnosis codes, with the most significant being $1 \times 10^{-7}$ for EAs (AUDIT-C: $\beta = 0.095$, $p = 8.98 \times 10^{-51}$; AUD: $\beta = 0.147$, $p = 6.02 \times 10^{-34}$) and $1 \times 10^{-7}$ for AAs (AUDIT-C: $\beta = 0.066$, $p = 3.69 \times 10^{-9}$; AUD: $\beta = 0.098$, $p = 1.09 \times 10^{-9}$).

Second, in an independent sample from the Penn Medicine BioBank, AUDIT-C and AUD PRS were significantly associated with alcohol-related disorders and alcoholism phecodes (see Methods and Supplementary Data 44 and 45). In EAs, higher AUDIT-C risk scores significantly increased the likelihood of alcohol-related disorders and alcoholism at multiple $p$ value thresholds, with the most significant being $1 \times 10^{-7}$ ($\beta = 0.278$, $p = 0.0013$) and $1 \times 10^{-6}$ ($\beta = 0.245$, $p = 0.0074$), respectively. In AAs, at a $p$ value threshold of $1 \times 10^{-7}$, AUDIT-C risk scores were non-significantly associated with risk of alcohol-related disorders ($\beta = 0.210$, $p = 0.064$) but significantly associated with alcoholism ($\beta = 0.400$, $p = 0.0051$). In both populations, AUD risk scores were significantly associated with both alcohol-related disorders and alcoholism. In EAs, the most significant $p$ value threshold was $1 \times 10^{-4}$ (alcohol-related disorders: $\beta = 0.254$, $p = 0.0006$; alcoholism: $\beta = 0.229$, $p = 0.0062$), while in AAs, the most significant $p$ value threshold was $1 \times 10^{-6}$ (alcohol-related disorders: $\beta = 0.306$, $p = 0.006$; alcoholism: $\beta = 0.440$, $p = 0.0007$).

Third, in the Yale-Penn study sample[25], an independent sample ascertained for substance use disorders, the PRS of AUDIT-C and AUD were significantly associated with DSM-IV alcohol dependence criterion counts (see Methods and Supplementary Data 46, 47). In EAs, all AUDIT-C risk scores were significantly associated with the criterion count, with the most significant $p$ value threshold being $1 \times 10^{-7}$ ($\beta = 1.029$, $p = 6.67 \times 10^{-13}$). Similarly, all AUD risk scores were significantly associated with the criterion count, the most significant $p$ value threshold being $1 \times 10^{-6}$ ($\beta = 1.144$, $p = 1.86 \times 10^{-16}$). In AAs, all but one AUDIT-C risk score and all AUD risk scores were significantly associated with the alcohol dependence criterion count, the most significant $p$ value threshold being $1 \times 10^{-7}$ (AUDIT-C: $\beta = 0.829$, $p = 1.18 \times 10^{-11}$; AUD: $\beta = 0.502$, $p = 4.62 \times 10^{-8}$).

**Secondary phenotypic associations**. To identify secondary phenotypes associated with AUDIT-C or AUD, we performed a phenome-wide association analysis (PheWAS) of the AUDIT-C and AUD PRS ($p$ value threshold = $1 \times 10^{-7}$ and all SNPs) in the MVP hold-out sample (Supplementary Data 48, 49). In EAs, the AUDIT-C PRS was significantly associated with an increased risk

of alcoholic liver damage, and nominally associated with a decreased risk of hyperglyceridemia. No significant associations were found for AAs. The AUD PRS was significantly associated with an increased risk of tobacco use disorder in both EAs and AAs, and in EAs with multiple psychiatric disorders, including major depression, bipolar disorder, anxiety, and schizophrenia.

## Discussion

We report here a GWAS of two alcohol-related traits in a sample of 274,424 MVP participants from five population groups—EA, AA, LA, EAA, and SAA—using two EHR-derived phenotypes: age-adjusted AUDIT-C score and AUD diagnostic codes. In addition to the large number of EAs, the study included large numbers of African-American and Latino-American participants. Trans-population meta-analyses identified 13 independent GWS loci for AUDIT-C and 10 independent GWS loci for AUD. For AUDIT-C, in addition to the loci identified in the SNP-based analyses, there were 31 GWS genes in EAs, 3 in AAs, 1 in LAs, and 2 in EAAs. For AUD, in addition to the loci identified in the SNP analyses, there were 23 GWS genes identified in EAs, 5 in AAs, and 1 in LAs.

Using both AUDIT-C scores and AUD diagnoses enabled us to examine the relations between these key alcohol-related traits. The findings underscore the utility of using an intermediate trait, such as alcohol consumption, for genetic discovery. Five of the 13 loci associated with AUDIT-C score, a measure of alcohol consumption, including the two most commonly identified alcohol metabolism genes (ADH1B and ADH1C) and three highly pleiotropic genes (GCKR, SLC39A8, and FTO), contributed to AUD risk. Of the 10 loci that were GWS for AUD, half also were associated with AUDIT-C score, while half were uniquely associated with the AUD diagnosis: ADH4, SIX3, a variant on chr10q25.1 and 2 variants in DRD2.

In addition to multiple overlapping variants for AUDIT-C and AUD, we found a moderate-to-high genetic correlation between the traits: 0.522 in EAs and 0.930 in AAs. There are two potential explanations for the population difference in genetic correlation. First, it may reflect a bias in the assignment of AUD diagnoses by clinicians (e.g., in the context of a high AUDIT-C score, clinicians could be less likely to assign an AUD diagnosis to EAs than AAs, reducing the genetic correlation). Second, because LD structure in admixed populations is complex, LD score regression could have inflated the genetic correlation among AAs, an admixed population. Another factor relevant to this difference is the smaller number of AAs, which despite a higher $r_g$, yielded a larger standard error. The genetic similarity between these alcohol-related traits is consistent with twin studies of alcohol dependence and alcohol consumption[31,32]. These findings are also consistent with the PRS analyses in the MVP sample, where both AUDIT-C and AUD PRS were associated with AUDIT-C and AUD phenotypes. Both traits also predicted multiple alcohol-related phenotypes in independent datasets, including alcohol dependence criteria in the Yale-Penn sample. However, there was a smaller effect of AUDIT-C PRS scores than AUD PRS scores on alcohol-related disorders and alcohol dependence. This is in line with findings from the meta-analysis of UKBB and 23andMe data, where the genetic correlation with alcohol dependence was nominally greater for AUDIT-P scores ($r_g = 0.63$) than AUDIT-C scores ($r_g = 0.33$)[8].

Despite the significant genetic overlap between the AUDIT-C and AUD diagnosis, downstream analyses revealed biologically meaningful points of divergence. The AUDIT-C yielded some GWS findings that did not overlap with those for AUD, which reflects genetic independence of the traits. This broadens our previous observations using SNPs in ADH1B, in which we

validated the AUDIT-C score as an alcohol-related phenotype[33]. In that study, after accounting for the effects of AUDIT-C score, AUD diagnoses accounted for unique variance in the frequency of ADH1B minor alleles.

Evidence of genetic independence between the two traits was most striking in the differences between the genetic correlation analyses. After correction, genetic correlations for 188 traits differed significantly (some in opposite directions) between AUDIT-C and AUD. Notably, these included a negative association of AUDIT-C with anthropometric traits, including BMI; coronary artery disease; and glycemic traits, including Type 2 diabetes. The negative genetic correlation with coronary artery disease is consistent with some epidemiological findings that alcohol consumption protects against some forms of cardiovascular disease[34]. AUDIT-C was positively genetically correlated with overall health rating, HDL cholesterol concentration, and years of education, findings that are consistent with prior literature showing genetic correlation of these traits with alcohol consumption[7,8,21]. AUD was significantly genetically correlated with 111 traits or diseases, including negative genetic correlations with intelligence, years of education and quitting smoking, and positive genetic correlations with insomnia, ever having smoked and most psychiatric disorders, findings that are consistent with phenotypic associations in the epidemiological literature[35–37] and genetic correlations reported from the UKBB and 23andMe GWASs and their meta-analysis[7,8,21]. The opposite genetic correlations seen for some traits may be driven by low-effect variants, as we find close to 100% consistency in the direction of effect for the most significantly associated SNPs for both AUDIT-C and AUD. Further, in the MVP sample, the AUD PRS was significantly positively associated with tobacco use and multiple psychiatric disorders, whereas the AUDIT-C PRS was not. Taken together, these findings suggest that AUD and alcohol consumption, measured by AUDIT-C, are related but distinct phenotypes, with AUD being more closely related to other psychiatric disorders, and AUDIT-C to some positive health outcomes.

Although the protective effects of moderate drinking are controversial, we found that alcohol consumption in the absence of genetic risk for AUD may protect from cardiovascular disease, diabetes mellitus, and major depressive disorder. In contrast, individuals with genetic risk for AUD are at elevated risk for some adverse secondary phenotypes, including insomnia, smoking, and other psychiatric disorders. However, individuals who have had health problems resulting from drinking are more likely to reduce or stop drinking by middle age or under-report their alcohol consumption. This offers an alternative explanation for the opposite genetic associations[38], particularly in an older clinical sample in which a large proportion report current abstinence (reflected in an AUDIT-C score of 0). For this complex set of genetic associations to be useful in informing clinical recommendations on safe levels of alcohol consumption, it will be necessary to elucidate the mechanisms underlying these findings.

Both phenotypes showed cell type-specific enrichments for CNS. Other relevant cell types for AUDIT-C, but not for AUD, included cardiovascular, adrenal or pancreas, liver, and musculoskeletal. Thus, although heavy drinking is prerequisite to the development of AUD, the latter is a polygenic disorder and variation in genes expressed in the CNS (e.g., DRD2) may be necessary for individuals who drink heavily to develop AUD. As a binary trait, AUD provided less statistical power to identify genetic variation than the ordinal AUDIT-C score, but the multiple GWS findings unique to AUD argue against that as an explanation for the non-overlapping GWS findings for the two traits.

The VHA EHR provided a rich source of phenotypic data. These included mean age-adjusted AUDIT-C scores, which are

more stable than measures at a single point in time (more likely reflecting traits rather than states) and contrast with meta-analytic studies that may use phenotypes reflecting the lowest-common denominator among the studies comprising the sample. However, our analyses were limited by our reliance on the AUDIT-C, which includes only the first 3 of the 10 AUDIT items. We also obtained cumulative AUD diagnoses, which are also more informative than assessments obtained at a single time point. Because the diagnosis of AUD is based on features other than alcohol consumption per se[2,5], use of the AUD diagnosis from the EHR augmented the information provided by the AUDIT-C phenotype. Although EHR diagnostic data are heterogeneous, large-scale biobanks such as the MVP yield greater statistical power to link genes to health-related traits repeatedly documented over time in the EHR than can ordinarily be achieved in prospective studies[23], justifying the lower resolution of EHR data. However, because the MVP sample is predominantly comprised of EA males, statistical power was limited in both the GWAS and the post-GWAS analyses of other populations and some female samples. Future studies with larger sample sizes are needed to identify additional variation contributing to these alcohol-related traits and to elucidate their interrelationship.

The SNP heritability of our GWASs was lower than that seen in the meta-analysis of the UKBB and 23andMe data[8]. For the AUDIT-C, the estimated SNP heritability was 0.068 in EAs (0.068 in males and 0.099 in females) and 0.062 in AAs. For AUD, the estimated SNP heritability was 0.056 in EAs (0.054 in males and 0.110 in females) and 0.100 in AAs. These estimates may reflect the lower number of SNPs tested in our sample compared with the meta-analysis of UKBB and 23andMe data. The nominally higher SNP heritability in females than males could be due to the substantially smaller size of the female subsample. Alternatively, women could have a higher liability-threshold and therefore a higher burden of risk variants. Because our study sample was predominantly male, we do not have adequate statistical power to evaluate these hypotheses. Although we found no significant difference in PRS between males and females, because of the substantially smaller number of women in MVP, there is much less power for the PRS in this subgroup and for comparing the PRS by sex.

Despite these limitations, the large, diverse, and similarly ascertained sample enabled us to identify multiple GWS findings for both AUDIT-C score and AUD diagnosis, and thereby to help elucidate the relationship between drinking level and AUD risk. The large sample provided high power for PRS analyses in other samples, as demonstrated here in the Penn Medicine Biobank and Yale-Penn samples. The genetic differences between the two alcohol-related traits and the observed opposite genetic correlations between them point to potentially important differences in comorbidity and prognosis. Our findings underscore the need to identify the functional effects of the risk variants, especially where they diverge by trait, to elucidate the nature of the trait-related differences. Focusing on variants linked to AUD, but not AUDIT-C, could identify targets for the development of medications to treat the disorder, while variation in AUDIT-C could help in developing interventions to reduce drinking and thereby prevent the morbidity associated with it. The findings reported here could also help to identify individuals at high risk of AUD through the use of PRS. This effort could be augmented using knowledge of the full set of phenotypes that associate with AUD through the use of genetic correlations and PheWASs.

## Methods

**Data collection**. The MVP is an observational cohort study and biobank supported by the U.S. Department of Veterans Affairs (VA). Phenotypic data were collected

from MVP participants using questionnaires and the VA EHR and a blood sample was obtained for genetic analysis.

Ethics statement: The Central Veterans Affairs Institutional Review Board (IRB) and site-specific IRBs approved the MVP study. All relevant ethical regulations for work with human subjects were followed in the conduct of the study and informed consent was obtained from all participants.

**Phenotypes**. AUDIT-C scores and AUD diagnostic codes were obtained from the VA EHR. The AUDIT-C comprises the first three items of the AUDIT and measures typical quantity (item 1) and frequency (item 2) of drinking and frequency of heavy or binge drinking (item 3). The AUDIT-C is a mandatory annual assessment for all veterans seen in primary care. Our analyses used AUDIT-C data collected from 1 October 2007 to 23 February 2017. We validated the phenotype in a sample of 1851 participants from the Veterans Aging Cohort Study[33], in which we found a highly significant association of AUDIT-C scores with the plasma concentration of phosphatidylethanol, a direct, quantitative biomarker that is correlated with the level of alcohol consumption. In the AA part of this sample ($n = 1503$), the AUDIT-C score was highly significantly associated with rs2066702, a missense (Arg369Cys) polymorphism of *ADH1B*, the minor allele of which is common in that population and has been associated with alcohol dependence[25]. We also examined AUDIT-C scores in 167,721 MVP participants (57,677 AAs and 110,044 EAs)[24], comparing the association of AUDIT-C scores and AUD diagnoses with the frequency of the minor allele of rs2066702 in AAs and rs1229984 (Arg48His) in EAs. Both polymorphisms exert large effects on alcohol metabolism[39] and are among the genetic variants associated most consistently with alcohol-related traits in both AAs and EAs[8,12,18]. In both populations, we found a stronger association between age-adjusted mean AUDIT-C score and *ADH1B* minor allele frequency than between AUD diagnostic codes and the frequency of the minor alleles[24]. However, because AUD diagnoses accounted for unique variance in the frequency of the minor alleles in both populations, we concluded that the two phenotypes, although correlated, are distinct traits. Thus, in the present study, we used GWAS to examine these traits separately and to adjust for the effects of AUD in the AUDIT-C GWAS and the effects of AUDIT-C in the GWAS of AUD.

We calculated the age-adjusted mean AUDIT-C value[24] for each participant using age 50 as the reference point and down-weighting scores for individuals younger than 50 and up-weighting scores for individuals older than 50. The age-adjusted mean AUDIT-C was computed using a sample of 495,178 participants with data on age and AUDIT-C, of whom 272,842 had genetic data and were included in the AUDIT-C genetic analyses.

The principal classes of alcohol-related disorders in the ICD are alcohol abuse and alcohol dependence. We used ICD-9 codes 303.X (dependence) and 305–305.03 (abuse) and ICD-10 codes F10.1 (abuse) and F10.2 (dependence) to identify subjects diagnosed with either of these disorders, as suggested previously[40] (see Supplementary Table 2). Participants with at least one inpatient or two outpatient alcohol-related ICD-9/10 codes ($N = 274,391$) from 2000 to 2018 were assigned a diagnosis of AUD, an approach that has been shown to yield greater specificity of ICD codes than chart review[41].

**Genotyping and imputation**. MVP GWAS genotyping was performed using an Affymetrix Axiom Biobank Array with 686,693 markers. Subjects or SNPs with genotype call rate <0.9 or high heterozygosity were removed, leaving 353,948 subjects and 657,459 SNPs for imputation[22].

Imputation was performed with EAGLE2 (ref. [42]) to pre-phase each chromosome and Minimac3 (ref. [43]) to impute genotypes with 1000 Genomes Project phase 3 data[44] as the reference panel. Subjects with no demographic information or whose genotypic and phenotypic sex did not match were removed. We also removed one subject randomly from each pair of related individuals (kinship coefficient threshold = 0.0884). A greedy algorithm was implemented for network-like relationships among three or more individuals, leaving 331,736 subjects for subsequent analyses.

**Population differentiation**. To differentiate population groups, we performed principal components analysis (PCA) using common SNPs (MAF > 0.05) shared in MVP [pruned using linkage disequilibrium (LD) of $r^2 > 0.2$] and the 1000 Genomes phase 3 reference panels for European (EUR), African (AFR), admixed American (AMR), East Asian (EAS), and South Asian (SAS) populations using FastPCA in EIGENSOFT[45]. We analyzed 80,871 SNPs in MVP and 1000 Genomes for use in the PCA analyses. The Euclidean distances between each participant and the centers of the five reference populations (i.e., across all subjects) were calculated using the first 10 PCs, with each participant assigned to the nearest reference population. A total of 242,317 EA; 61,762 AA; 15,864 Hispanic and Latino American (LA); 1565 East Asian American (EAA); and 228 South Asian American (SAA) subjects were identified. A second PCA (within each group) yielded the first 10 PCs for each. Participants with PC scores >3 standard deviations from the mean of any of the 10 PCs were removed as outliers, leaving 209,020 EA; 57,340 AA; 14,425 LA; 1410 EAA; and 196 SAA subjects. Within genetically defined populations, we calculated population-specific imputation INFO scores using SNPTEST v2 (ref. [46]) and retained SNPs with INFO scores >0.7 for association analyses.

Imputed genotypes with posterior probability ≥0.9 were transferred to best guess. We removed both genotyped and imputed SNPs with genotype call rates or best guess rates ≤0.95 and HWE $p$ value ≤ $1 \times 10^{-6}$ in each population, using different MAF thresholds to filter SNPs: EA (0.0005), AA (0.001), LA (0.01), EAA (0.05), and SAA (0.05). The approximate number of SNPs remaining in each population was EA: 6.8 million, AA: 12.5 million, LA: 5.6 million, EAA: 2.6 million, and SA: 2.6 million.

**Genome-wide association analyses.** Individuals <22 or > 90 years old and those with missing AUDIT-C scores were removed from the analyses, leaving 200,680 EAs; 56,495 AAs; 14,112 LAs; 1366 EAAs; and 189 SAAs in the AUDIT-C GWAS and 202,004 EAs (34,658 cases; 167,346 controls); 56,648 AAs (17,267 cases; 39,381 controls); 14,175 LAs (3449 cases; 10,726 controls); 1374 EAAs (164 cases; 1210 controls); and 190 SAs (44 cases; 144 controls) in the AUD GWAS. We used linear regression for the GWAS of age-adjusted mean AUDIT-C score and logistic regression for AUD diagnosis; in both cases age, sex, and the first 10 PCs were covariates. To evaluate the impact on AUD findings of controlling for AUDIT-C and the impact on AUDIT-C findings of controlling for AUD, we repeated the GWAS for AUD with AUDIT-C as a covariate and AUDIT-C with AUD as a covariate. For both phenotypes, following GWAS in each of the five populations, the summary statistics were combined within phenotype in trans-population meta-analyses. SNPs in EAs or those present in at least two populations were meta-analyzed. Sex-stratified GWAS for both phenotypes were then performed in groups large enough to permit it—EA, AA, LA, and EAA men and EA, AA, and LA women—and the data were meta-analyzed within sex and phenotype. All meta-analyses were performed using a sample-size-weighted scheme that was implemented in METAL[47].

To identify independent signals in each population, we performed LD clumping using PLINK v1.90b4.4 (ref. [48]). We identified an index SNP ($p < 5 \times 10^{-8}$) with the smallest $p$ value in a 500-kb genomic window and $r^2 < 0.1$ with other index SNPs. Because in EAAs there is extended linkage disequilibrium at the *ALDH2* locus, we used a 2500-kb window in that population. In the chr4q23–q24 region, where we identified multiple apparently independent signals for both AUDIT-C and AUD, we used conditional associations to differentiate independent signals from partially overlapping ones.

**Gene-based association analysis.** Gene-based association analysis was performed using Multi-marker Analysis of GenoMic Annotation (MAGMA)[49], which uses a multiple regression approach to detect multi-marker effects that account for SNP $p$ values and LD between markers. We used the default setting (no window around genes) to consider 18,575 autosomal genes for the analysis, with $p < 2.69 \times 10^{-6}$ (0.05/18,575) considered GWS. For each population, we used the respective population from the 1000 Genomes Project phase 3 as the LD reference.

**Enrichment analyses.** Pathway and biological enrichment analyses were performed for each population using the FUMA platform[26], with independent significant SNPs identified using the default settings. Positional gene mapping identified genes up to 10 kb from each independent significant SNP. Hypergeometric tests were used to examine the enrichment of prioritized chemical and genetic perturbation gene sets, canonical pathways, and GO biological processes (obtained from MsigDB c2), and GWAS-catalog enrichment (obtained from reported genes from the GWAS-catalog). We report all significantly enriched gene sets based on an FDR-adjusted $p$ value <0.05.

**Heritability and partitioning of heritability.** LDSC[27] was used to calculate population-specific LD scores based on 1000 Genomes phase 3 datasets according to the LDSC tutorial, using SNPs selected from HapMap 3 (ref. [50]) after excluding the major histocompatibility complex (MHC) region (chr6: 26–34Mb); only ancestry groups with large sample size ($N > 10,000$) were analyzed using LDSC. Of note, LDSC could be biased in admixed populations because reference panels are not provided for AAs and LAs in that application[27]. We calculated LD scores for 1,215,001 SNPs in EAs; 1,322,841 SNPs in AAs; and 1,243,726 SNPs in LAs. The LDSC analyses used SNPs with imputation INFO ≥ 0.9 in each population and that were LD scored in 1000 Genomes. LD score regression intercepts for available datasets were estimated to distinguish polygenic heritability from inflation. SNP-based heritability ($h_{2\,SNP}$) was estimated from GWAS summary statistics for both AUDIT-C and AUD. The sex-specific $h_{2\,SNP}$ was also estimated in EA males and females, AA males, and LA males.

We estimated partitioned $h_{2\,SNP}$ using genomic features or functional categories[28] for both AUDIT-C and AUD in the largest dataset, EAs, and then tested for enrichment of the partitioned $h_{2\,SNP}$ in different annotations. First, we used a baseline model consisting of 53 functional categories, including UCSC gene models [exons, introns, promotors, untranslated regions (UTRs)], ENCODE functional annotations[51], Roadmap epigenomic annotations[52], and FANTOM5 enhancers[53]. We then analyzed cell type-specific annotations and identified enrichments of $h_{2\,SNP}$ in 10 cell types, including adrenal and pancreas, CNS, cardiovascular, connective tissue and bone, gastrointestinal, immune and hematopoietic, kidney, liver, skeletal muscle and other. Gene expression and chromatin data were also analyzed to identify disease-relevant tissues, cell types,

and tissue-specific epigenetic annotations. We used LDSC to test for enriched heritability in regions surrounding genes with the highest tissue-specific expression or with epigenetic marks[29]. Sources of data that were analyzed included 53 human tissue or cell type RNA-seq data from the Genotype-Tissue Expression Project (GTEx)[54]; 152 human, mouse, or rat tissue or cell type array data from the Franke lab;[55] 3 sets of mouse brain cell type array data from Cahoy et al.[56]; 292 mouse immune cell type array data from ImmGen[57]; and 396 human epigenetic annotations (6 features in 88 primary cell types or tissues) from the Roadmap Epigenomics Consortium[52]. In the analysis of each trait in each dataset, we used FDR < 0.05 to indicate significant enrichment for the $h_{2\,SNP}$.

**Genetic correlations.** We estimated the genetic correlation ($r_g$) between AUDIT-C and AUD (from MVP), and with other traits in LD Hub[58] or from published studies using LDSC, which is robust to sample overlap[30]. First, we estimated the $r_g$ between AUDIT-C and AUD using the summary data generated in this study. AAs, EAs, and LAs were analyzed separately using the corresponding 1000 Genome phase 3 population as reference. Genetic correlations in EAA and SAA were not analyzed. The $r_g$ between AUDIT-C and AUD was out of bounds because the $h_{2\,SNP}$ did not differ from zero. Then we tested the $r_g$ between males and females within each trait. We estimated the $r_g$ for AUDIT-C and AUD with 216 published traits in LD Hub and 493 unpublished traits from the UK Biobank. We resolved redundancy in phenotypes by manually selecting the published version of the phenotype or using the largest sample size. We also calculated the genetic correlations of both AUDIT-C and AUD with five traits for which GWAS were recently published or posted, including anorexia nervosa[59], alcohol dependence[13], attention deficit hyperactivity disorder[60], autism spectrum disorder[61], and major depressive disorder (summary data without the 23andMe sample)[62], bringing the total number of tested traits to 714. A Bonferroni correction was applied separately for AUDIT-C and AUD, and traits with a corrected $p$ value <0.05 were considered significantly correlated. Because the results were similar whether the intercept was constrained or not, we present here the original results without constraint.

**Polygenic Risk Scores.** To generate PRS from GWAS summary statistics in the MVP sample, we first conducted a GWAS for AUDIT-C and AUD as above, but restricted our analysis to two-thirds of the total sample by splitting the total sample randomly, keeping the number of AUD cases/controls balanced in each part (EA: $N = 139,346$, AA: $N = 38,226$). GWS loci identified from this analysis were the same as those in the larger sample, but were slightly decreased in significance. PRS were generated for the remaining EAs ($N = 69,674$) and AAs ($N = 19,114$) as the sum of all variants carried, weighted by the effect size of the variant in the GWAS. PRS were generated using PLINK2 (ref. [63]). We performed $p$ value informed clumping with a distance threshold of 250 kb and $r^2 = 0.1$. Risk scores were calculated for a range of $p$ value thresholds ($P \le 1 \times 10^{-7}$, $1 \times 10^{-6}$, $1 \times 10^{-5}$, $1 \times 10^{-4}$, $1 \times 10^{-3}$, 0.01, 0.05, 0.5, 1.0). PRS were standardized with mean = 0 and SD = 1. Logistic regression was used to test for association with AUDIT-C and AUD phenotypes, with PRS as the independent variable and AUDIT-C or AUD as the dependent variable, with age, sex, and the first five PCs as covariates.

Population-specific summary statistics from the AUDIT-C and AUD GWAS in MVP were used to generate PRS in the PMBB, an independent sample. PRS were generated for EAs ($N = 8524$) and AAs ($N = 2031$) as above using the PRSice2 package[64] with imputed allele dosage as the target dataset. As recommended in the software, we performed $p$ value informed clumping with a distance threshold of 250 kb and $r^2 = 0.1$. We excluded the MHC region. Risk scores were calculated for a range of $p$ value thresholds ($p \le 1 \times 10^{-7}$, $1 \times 10^{-6}$, $1 \times 10^{-5}$, $1 \times 10^{-4}$, $1 \times 10^{-3}$, 0.01, 0.05, 0.5, 1.0) and standardized with mean = 0 and SD = 1. To identify individuals with alcohol-related disorders, we utilized phecodes, a method to aggregate ICD codes[65]. First, we extracted ICD-9 and ICD-10 data for 48,610 individuals from the EHR. To facilitate mapping to phecodes, ICD-10 codes were back converted to ICD-9 using 2017 general equivalency mapping (GEM). The ICD-10 conversions were combined with the ICD-9 codes to create a dataset with 10,682 unique ICD-9 codes. ICD-9 codes were aggregated to phecodes using the PheWAS R package[65] to create 1812 phecodes. Individuals are considered cases for the phenotype if they had at least two instances of the phecode, controls if they had no instance of the phecode, and other/missing if they had one instance or a related phecode. Logistic regression was used to test the association of the PRS with the alcohol-related disorders phecode (phecode number 317) and its sub-phenotype, alcoholism (phecode number 317.1). The analysis was performed in R with PRS as the independent variable and diagnosis as the dependent variable and age, sex, and the first 10 PCs as covariates.

We also tested the PRS of AUDIT-C and AUD for DSM-IV alcohol dependence criterion counts in the Yale-Penn cohort[25]. There are three phases of the Yale-Penn sample: phase 1 contains 3110 AAs and 1718 EAs exposed to alcohol; phase 2 contains 1667 AAs and 1689 EAs exposed to alcohol; phase 3 contains 556 AAs and 999 EAs exposed to alcohol. PRS were generated for EAs and AAs in each phase as described above and risk scores were calculated for a range of $p$ value thresholds ($p \le 1 \times 10^{-7}$, $1 \times 10^{-6}$, $1 \times 10^{-5}$, $1 \times 10^{-4}$, $1 \times 10^{-3}$, 0.01, 0.05, 0.5, 1.0). Different from PRS in MVP and PMBB, to correct for the relatedness in the Yale-Penn subjects, a linear mixed model implemented in GEMMA[66] was used to test the association between PRS score and DSM-IV alcohol dependence criterion counts,

with age, sex, and the first 10 PCs as covariates. Meta-analyses of data from the three phases were performed in AAs ($N = 5333$) and EAs ($N = 4406$) separately.

**Phenome-wide association analysis**. To conduct PheWAS, we extracted ICD-9 data from the EHR for 353,323 genotyped veterans. Of these, 277,531 individuals had two or more separate encounters in the VA Healthcare System in each of the 2 years prior to enrollment in MVP, consisting of 21,209,658 records. ICD-9 codes were aggregated to phecodes using the PheWAS R package to create 1812 phecodes. To improve the specificity of these codes, individuals with at least two instances of the phecode were considered cases, those with no instance of the phecode controls, and those with one instance of a phecode or a related phecode as other. A PheWAS using logistic regression models with either AUDIT-C or AUD PRS as the independent variables, phecodes as the dependent variables, and age, sex and the first five PCs as covariates were used to identify secondary phenotypic associations. A phenome-wide significance threshold of $2.96 \times 10^{-5}$ was applied to account for multiple testing.

Secondary GWAS Adjusted for BMI: As described below, for both alcohol-related traits, we identified a GWS SNP in *FTO*, variation in which has been associated with BMI and risk of obesity[67]. To examine whether BMI confounded the association with this and other loci and the genetic correlations with other traits, we repeated the GWAS for AUDIT-C and AUD using BMI as an additional covariate. Data on BMI were from the MVP baseline survey and the EHR. For AUDIT-C, 200,092 EAs; 56,239 AAs; 14,029 LAs; 1352 EAAs; and 185 SAAs had BMI data available. For AUD, 201,320 EAs; 56,347 AAs; 14,075 LAs; 1360 EAAs; and 186 SAAs had BMI data available. After GWAS, we analyzed the genetic correlations between BMI-adjusted traits and other publicly available traits ($N = 714$), with Bonferroni correction for multiple testing.

**Reporting summary**. Further information on experimental design is available in the Nature Research Reporting Summary linked to this article.

## Data availability
The full summary-level association data from the meta-analysis for each of the two alcohol-related traits from this report are available through dbGaP: [https://www.ncbi.nlm.nih.gov/projects/gap/cgi-bin/study.cgi?study_id=phs001672.v1.p1] (accession number phs001672.v1.p1). Further information on research design is available in the Nature Research Reporting Summary linked to this article. All other data are contained within the article and its supplementary information are available upon reasonable request from the corresponding author.

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

## Acknowledgements

This research is based on data from the Million Veteran Program (MVP), Office of Research and Development, Veterans Health Administration, and was supported by award #I01BX003341. This publication does not represent the views of the Department of Veterans Affairs or the United States Government. A full acknowledgment of the MVP is included in Supplementary Note 1. We also appreciate access to summary data provided by the Psychiatric Genomics Consortium (PGC) Substance Use Disorders (SUD) working group. The PGC-SUD is supported by funds from NIDA and NIMH to MH109532 and, previously, had analyst support from NIAAA to U01AA008401 (COGA). PGC-SUD gratefully acknowledges its contributing studies and the participants in those studies without whom this effort would not be possible. Supported by the Mental Illness Research, Education and Clinical Center of the Veterans Integrated Service Network 4 of the Department of Veterans Affairs.

## Author contributions

H.R.K., J.G., A.C.J., J.C., R.L.K. and H. Zhao designed the study. H.R.K., J.G., A.C.J. and H. Zhao supervised the work. H. Zhou, R.L.K. and R.V.S conducted the analyses. S.D., P.S.T., D.K. and D.J.R. provided phenotypic data and the Regeneron Genetics Center provided genotypic data for the phenome-wide association analyses. The manuscript was written by H.R.K., H. Zhou, R.L.K., R.V.S. and J.G., with comments provided by all other authors. All authors approved the final version.

## Additional information

**Competing interests:** H.R.K. is a member of the American Society of Clinical Psychopharmacology's Alcohol Clinical Trials Initiative, which in the past three years was supported by AbbVie, Alkermes, Ethypharm, Indivior, Lilly, Lundbeck, Otsuka, Pfizer, Arbor, and Amygdala Neurosciences. H.R.K. and J.G. are named as inventors on PCT patent application #15/878,640 entitled: "Genotype-guided dosing of opioid agonists," filed 24 January 2018. The remaining authors declare no competing interests.

