## [Peer Review File · Nature Communications]

Reviewer #1 (Remarks to the Author):

The authors have put a lot of work into the revised version of the manuscript. Especially the PRS analyses have become much more convincing, and in general the quality of the work has improved considerably. I think the authors have responded well to my comments and I only have some minor points:

Page 13, line 357, the authors write: "In addition to multiple overlapping variants for AUDIT-C and AUD, we found a moderate-to-high genetic correlation between the traits: 0.522 in EAs and 0.930 in AAs. This population difference may reflect a bias in the assignment of AUD diagnoses by clinicians (e.g., in the context of a high AUDIT-C score, clinicians could be less likely to assign an AUD diagnosis to EAs than AAs)."

I do not follow the argument. If this is happening in the clinic I think it would increase r_g rather than decreasing it for EA. If individuals with a high AUDIT-C score have a higher tendency to be put in the AUD-C group rather than the AUD group (compared to AA) it would make the AUD-C group genetically more similar to the AUD group, as far as I see it.

I appreciate the sign test the authors have included in the manuscript, where they find close to 100% consistency of the direction of association of the most associated variants with AUD-C and AUD. Does this mean that the opposite genetic correlations seen for some traits (e.g. educational attainment), is driven by low effect size variants? I don't think this needs to be tested, but maybe a sentence in the discussion.

The authors write page 16 line 438: "The nominally higher SNP heritability in females than males may be due to the substantially smaller size of the female subsample."

Following the liability threshold model could it be that women (as the rare sex with the disorder), have a higher threshold (and thus higher burden of risk variants)?

Reviewer #3 (Remarks to the Author):

I've been one of the reviewers of the original submission of this paper. I'm satisfied with the way the authors handled the review comments and suggestions. The manuscript clearly improved by taking all reviewers suggestions into account. I have no further comments.

RESPONSE TO REVIEWERS' COMMENTS:

Reviewer #1

--The authors have put a lot of work into the revised version of the manuscript. Especially the PRS analyses have become much more convincing, and in general the quality of the work has improved considerably. I think the authors have responded well to my comments and I only have some minor points.

Response: We thank the reviewer for the many constructive suggestions that he or she provided, which we agree have led to a substantially improved manuscript.

--Page 13, line 357, the authors write: "In addition to multiple overlapping variants for AUDIT-C and AUD, we found a moderate-to-high genetic correlation between the traits: 0.522 in EAs and 0.930 in AAs. This population difference may reflect a bias in the assignment of AUD diagnoses by clinicians (e.g., in the context of a high AUDIT-C score, clinicians could be less likely to assign an AUD diagnosis to EAs than AAs)."

I do not follow the argument. If this is happening in the clinic I think it would increase r_g rather than decreasing it for EA. If individuals with a high AUDIT-C score have a higher tendency to be put in the AUD-C group rather than the AUD group (compared to AA) it would make the AUD-C group genetically more similar to the AUD group, as far as I see it.

Response: In EAs, false negative cases of AUD with high AUDIT-C scores would contribute error to the genetic correlation by showing that, for example, the genetic variation associated with high AUDIT-C scores is associated with the absence of an AUD diagnosis. This may also have reduced the genetic correlation between AUD and AUDIT-C in EAs. In contrast, among AAs, those with both high AUDIT-C scores and AUD (true positives) would share more genetic variation, yielding a higher genetic correlation in this population. We added another possible explanation for the difference in genetic correlation between the two population groups: namely, we acknowledged that, because LD structure in admixed populations is complex, LD score regression could have inflated the genetic correlation among AAs, an admixed population.

--I appreciate the sign test the authors have included in the manuscript, where they find close to 100% consistency of the direction of association of the most associated variants with AUD-C and AUD. Does this mean that the opposite genetic correlations seen for some traits (e.g. educational attainment), is driven by low effect size variants? I don't think this needs to be tested, but maybe a sentence in the discussion.

Response: We added a sentence to the discussion as suggested, i.e., that low-effect variants are potential contributors to the opposite genetic correlations that we observed for some traits in relation to AUDIT-C and AUD.

--The authors write page 16 line 438: "The nominally higher SNP heritability in females than males may be due to the substantially smaller size of the female subsample."

Following the liability threshold model could it be that women (as the rare sex with the disorder), have a higher threshold (and thus higher burden of risk variants)?

Response: We revised the sentence identified by the reviewer to include the potential that women have a higher threshold and a higher burden of risk variants. We also acknowledged that the preponderantly male study sample does not provide adequate statistical power to test these hypothesized explanations.

Reviewer #3 (Remarks to the Author):

I've been one of the reviewers of the original submission of this paper. I'm satisfied with the way the authors handled the review comments and suggestions. The manuscript clearly improved by taking all reviewers suggestions into account. I have no further comments.

Response: We thank the reviewer for the constructive suggestions that he or she provided, which led to a substantially improved manuscript.